# SUMOylation Protects FASN Against Proteasomal Degradation in Breast Cancer Cells Treated with Grape Leaf Extract

**DOI:** 10.3390/biom10040529

**Published:** 2020-03-31

**Authors:** Andrea Floris, Michael Mazarei, Xi Yang, Aaron Elias Robinson, Jennifer Zhou, Antonio Barberis, Guy D’hallewin, Emanuela Azara, Ylenia Spissu, Ainhoa Iglesias-Ara, Sandro Orrù, Maria Lauda Tomasi

**Affiliations:** 1Department of Medicine, Los Angeles, CA 90048, USA; andreafloris1179@gmail.com (A.F.); michael.mazarei@cshs.org (M.M.); xi.yang@cshs.org (X.Y.); j.zhou4197@yahoo.com (J.Z.); 2Cedars-Sinai Medical Center, Los Angeles, CA 90048, USA; aaron.robinson@gmail.com; 3Department of Medicine, Advanced Clinical Biosystems Research Institute, The Smidt Heart Institute, Los Angeles, CA 90048, USA; 4Institute of Sciences of Food Production (ISPA), 07100 Sassari, Italy; antonio.barberis@cnr.it (A.B.); guy.dhallewin@ispa.cnr.it (G.D.); yspissu@uniss.it (Y.S.); 5National Research Council (CNR), 07100 Sassari, Italy; emanuela.azara@icb.cnr.it; 6Institute of Biomolecular Chemistry (ICB), 07100 Sassari, Italy; 7Department of Genetics, Physical Anthropology and Animal Physiology, University of the Basque Country (UPV/EHU), 48940 Bilbao, Spain; ainhoa.iglesias@ehu.es; 8Department of Medical Sciences and Public Health, University of Cagliari, 09042 Cagliari, Italy; s.orru@unica.it

**Keywords:** antioxidant, breast cancer, FASN, lipid metabolism, polyphenols, protein degradation, protein stability, ubiquitination, SUMOylation

## Abstract

Existing therapeutic strategies for breast cancer are limited by tumor recurrence and drug-resistance. Antioxidant plant-derived compounds such as flavonoids reduce adverse outcomes and have been identified as a potential source of antineoplastic agent with less undesirable side effects. Here, we describe the novel regulation of fatty-acid synthase (FASN), the key enzyme in de novo fatty-acid synthesis, whereby *Vitis vinifera* L. cv Vermentino leaf hydroalcoholic extract lowers its protein stability that is regulated by small ubiquitin-like modifier (SUMO)ylation. The phenolic compounds characterization was performed by liquid chromatography–mass spectrometry (LC–MS), whereas mass spectrometry (LC–MS/MS), Western blotting/co-immunoprecipitation (Co-IP) and RT-PCR, 3-(4,5-dimethylthiazol-2-yl)-2,5-diphenyltetrazolium bromide (MTT), clonogenicity assays, and FACS analysis were used to measure the expression of targets and tumorigenicity. Vermentino extract exhibits antitumorigenic effects, and we went on to determine that FASN and ubiquitin-conjugating enzyme 9 (UBC9), the sole E2 enzyme required for SUMOylation, were significantly reduced. Moreover, FASN was found SUMOylated in human breast cancer tissues and cell lines, and lack of SUMOylation caused by SUMO2 silencing reduced FASN protein stability. These results suggest that SUMOylation protects FASN against proteasomal degradation and may exert oncogenic activity through alteration of lipid metabolism, whereas Vermentino extract inhibits these effects which supports the additional validation of the therapeutic value of this compound in breast cancer.

## 1. Background

Breast cancer is the most common malignant cancer in females worldwide [1]. The existing therapeutic strategies for breast cancer, which include surgery, endocrine therapy, and chemotherapy, are limited by tumor recurrence and drug resistance [2]. Therefore, novel approaches are needed to enhance the efficacy of existing therapeutic agents and to improve current clinical protocols.

Adjuvant therapies often attempt to induce cytotoxicity in tumor cells. As tumor cells are known to rely on alternate metabolic processes, such as de novo fatty-acid synthesis, these pathways harbor many potential therapeutic targets. In fact, inhibition of fatty-acid synthesis promotes apoptosis and produces cytotoxicity, which can trigger cell death [3]. A key enzyme in de novo fatty-acid synthesis is fatty-acid synthase (FASN). FASN catalyzes acetyl coenzyme A (CoA) and malonyl-Co to form palmitate and a 16-carbon fatty acid [4]. FASN is highly expressed in various breast cancer cell lines, including hormone-independent lines, such as SKBR-3, and hormone dependent lines, such as MCF-7 [5]. FASN can be regulated through genetic modulation and/or nuclear maturation of an isoform of SREBF1 (sterol regulatory element binding transcription factor 1), SREBP1c (sterol regulatory element binding protein 1c). SREBP1c is a transcription factor that binds the FASN promoter and increases the transcription rate of FASN [6]. In addition, FASN positively regulates and is regulated by expression of AKT serine/threonine kinase 1 (AKT1). AKT1 activation protects cells from cell death following inhibition of fatty-acid synthesis [7].

Another attractive addition to current clinical protocols is recommendation of a diet high in naturally occurring antioxidants. High consumption of fruits and vegetables containing antioxidative vitamins, carotenoids, and others small molecules with chemo-preventative activity can be an important element of primary cancer prevention [8]. Plant pharmacological activity is strongly correlated with natural antioxidants [9]. Plant-derived antioxidant compounds, such as flavonoids, reduce adverse outcomes of reactive oxygen and nitrogen species and are a potential source of antineoplastic and cytotoxic agents with fewer undesirable side effects [10,11]. Several epidemiological studies indicate that high flavonoid intake is correlated with reduced risk of cancer [12]. In vitro studies demonstrate that several mechanisms are linked to flavonoid-mediated cytotoxicity, including cell proliferation inhibition, adhesion, invasion, cell differentiation with simultaneous cell cycle arrest inhibition, and apoptosis [13,14]. Flavonoids are a sub-group of more than 5000 polyphenolic compounds with excellent antioxidant properties that are naturally produced in considerable quantities in fruits and leaves [15]. Oxidation to ortho- and para-quinones are the standard aromatic transformation related to the presence of the phenolic hydroxyl, which contributes to antioxidant characteristics [16].

Here, we investigated *Vitis vinifera* L. cv. Vermentino because it is representative of the productive activities of the Sardinian territory and has a unique phenolic profile compared to other vines commonly produced in north Italy. The aim of this study was to explore the potential cytotoxicity of the hydroalcoholic extract of Vermentino leaves as well as the cell signaling pathways implicated in any potential pharmacological effects.

## 2. Methods

### 2.1. Cell Culture and Treatments

The human breast cancer cell lines MCF-7 and SKBR-3 as well as the MCF-12A normal breast epithelial cell line were obtained from the American Type Culture Collection (ATCC, Rockville, MD, USA) and were grown according to instructions provided by the ATCC. These epithelial cell lines were routinely cultured in high glucose Dulbecco’s Modified Eagle Medium (DMEM) containing 4.5 g/L D-glucose and supplemented with 10% fetal bovine serum and penicillin (100 U/mL)/streptomycin (100 U/mL) at 37 °C under 5% CO_2_. In this study, cells were exposed to Vermentino leaf hydroalcoholic extract for 16 and 24 h at concentrations of 100, 200, and 400 µg/mL. In the experiments described below, cells were treated with 60 μg/mL cycloheximide (CHX) or 0.5 μM MG132 (carbobenzoxy-Leu-Leu-leucinal) (Sigma-Aldrich, St. Louis, MO, USA) for 4, 18, and 24 h. The medium was changed after 15 min of MG132 pretreatment.

### 2.2. Human Breast Tissue Specimens

Three normal breast tissues and five breast cancer tissues from surgical reductive mammoplasty and surgical resection of primary breast cancer, respectively, were used (Appendix A). All tissues were immediately frozen in liquid nitrogen for subsequent protein extraction. Written informed consent was obtained from each patient. The study protocol conformed to the ethical guidelines of the 1975 Declaration of Helsinki, as reflected by a prior approval of the study by Cedars-Sinai Medical Center’s human research review committee.

### 2.3. Plant Material and Extraction Procedure

Mature leaves of *Vitis vinifera* L. cultivar Vermentino were collected in August from the apical portion of plants cultivated in the collection field of the Institute of Sciences of Food Production (ISPA) in Oristano, Italy. The leaves were immediately put in a portable fridge and cooled to 4 °C to be moved to the ISPA laboratories. There, the leaves were rinsed with tap water, ground into a fine powder in liquid nitrogen to avoid the degradation of thermolabile compounds, and processed for polyphenol extraction.

An accelerated solvent extraction was performed following a previously established methodology [17], with some modifications. An hydroalcoholic (ethanol/water 40%/60%) solvent was used for the polyphenol extraction from leaf powder. It was performed in triplicate. The leaf powder was put in contact with the hydroalcoholic solvent for 16 h, thus obtaining a suspension. The suspension was then centrifuged at 4629 *g* for 10 min at 4 °C (A.L.C.-4227R, A.L.C. s.r.l., Milan, Italy). Only the supernatant, the hydroalcoholic extract containing the polyphenolic fraction, was frozen at −80 °C and subjected to gaseous nitrogen flow to remove ethanol. Then, it was lyophilized to eliminate all the water and any ethanol residues. The result was a polyphenolic ethanol-free extract powder used for analyses and treatments. This extract was suitably diluted in the culture medium to carry out cell treatments.

The yield of extraction was calculated as the ratio between “the weight of freeze-dried recover” and “the initial weight of leaf powder used” and expressed as a percentage.

### 2.4. Phenolic Characterization of Grapevine Extract

#### 2.4.1. HPLC-UV Analysis

Vermentino extract was filtered through 0.2 μm regenerate cellulose membrane syringe filters (Phenomenex, Torrence, CA, USA). Phenolic compounds were determined by LC–MS (liquid chromatography–mass spectrometry), according to previously described conditions [18]. A DAD (diode array HPLC detector) was used at 280, 320, and 520 nm for quantitative analyses. The quantification of phenolic compounds was performed using the external calibration curves according to commercial standards; the quercetin 3-O-(6 acetyl) glucoside content was calculated using a quercetin 3-O glucoside standard curve.

#### 2.4.2. LC-HRMS Analysis

High resolution (HR) MS analyses were performed on a QExactive Orbitrap (Thermo Scientific, Bremen, Germany) coupled to 1200 series HPLC (Agilent Technologies, Santa Clarita, CA, USA) equipped with a binary pump, a thermostatic autosampler, and a column oven set to 39 °C.

To investigate the secondary metabolite profile, the QExactive was equipped with a heated electrospray ionization source (HESI), operating in both positive and negative ion mode. The HESI parameters were spray voltage, 3.2 kV; sheath gas flow rate, 35 (arbitrary units); auxiliary gas, 10 (arbitrary units); sweep gas, 2 (arbitrary units); and capillary temperature, 300 °C. Full MS acquisition was performed with a resolution power of 70,000 full width at half maximum (FWHM) for parent ions and 17,500 for the fragment ions with mass accuracy of 5 ppm. The MS parameters were Automatic Gain Control target 1e6, maximum injection time (IT) 200 ms, and scan range 100–1200 m/z. The Xcalibur 3.1.66 software (Thermo Scientific, Bremen, Germany) was used to control the instruments and to process the data. A Gemini C18 (Phenomenex, Torrance, CA, USA; 100 × 2.1 mm, 3 μm, 100Å) was used for chromatographic separation. The flow rate was 0.2 mL/min during a 55 min period with an injection volume of 5 μL. A linear gradient elution of solvent acetic acid 0.2% (A) and acetonitrile (B) was applied with the following program: 0 min, 10% B; 0–20 min, 10–20% B; 20–40 min, 20–40% B; and 40–50 min, 40–70% B. The column was equilibrated for 8 min prior to each analysis. These conditions were adapted from our previous study [18].

Peaks were identified on the basis of their retention time relative to external standards (tR), UV-VIS spectra (200–650 nm), high resolution mass spectra, phytochemical libraries, and reference literature. Quantification of the single phenolic compound was performed using calibration curves of the respective reference compound. When reference compound was not available, the calibration was based on structurally related molecules.

### 2.5. Western Blot and Co-Immunoprecipitation (Co-IP)

MCF-7 and SKBR-3 cells were washed with cold phosphate-buffered saline (PBS), and then total proteins were extracted with 100 uL of radioimmunoprecipitation assay (RIPA) buffer (20 mM Tris-HCl (pH 7.5), 150 mM NaCl, 1 mM Na_2_EDTA, 1 mM EGTA, 1% NP-40, 1% sodium deoxycholate) containing 1X protease inhibitors cocktail from Sigma (cat.# P8340; (St. Louis, MO, USA). Immunoprecipitation by specific antibodies was performed as previously reported [19]. Immunoprecipitated proteins were subjected to Western blotting (WB) following standard protocols (Amersham BioSciences, Piscataway, NJ, USA), and the membranes were probed for FASN, AKT1, p-AKT1, small ubiquitin-like modifier (SUMO)1, SUMO2/3, ubiquitin-conjugating enzyme 9 (UBC9) (Santa Cruz Biotechnology, Dallas, TX, USA), Ubiquitin (UBC) (Proteintech, Rosemont, IL, USA), caspase-3 (CASP-3), caspase-9 (CASP-9), and β-actin (Sigma-Aldrich). Bands were detected using an enhanced chemiluminescence detection system (Millipore Corporation, Billerica, MA, USA). Quantification of relative band intensity was performed using Quantity One version 4.6.6. basic (Bio-Rad, Hercules, USA), and β-actin was used as a normalizing factor.

### 2.6. Real-Time PCR (RT-PCR) Analysis

Total RNA was extracted using a Quick-RNA Kit (Zymo Research, Irvine, CA, USA), according to the manufacturer’s protocol. The first strand of cDNA was synthesized via reverse transcription by M-MLV reverse transcriptase (Invitrogen, Carlsbad, CA, USA). Quantitative real-time PCR analysis was performed using 2 µL of PCR product. TaqMan probes for human FASN (fatty-acid synthase) and the Universal PCR Master Mix were purchased from ABI (Foster City, CA, USA). Hypoxanthine phosphoribosyl-transferase 1 (HPRT1) was used as a housekeeping gene. The delta Ct (ΔCt) obtained was used to find the relative expression of genes, according to the following formula: relative expression = 2–ΔΔCt, where ΔΔCt = ΔCt of respective genes in experimental groups – ΔCt of the same genes in control group.

### 2.7. Cell Viability

The MTT assay (3-(4,5-dimethylthiazol-2-yl)-2,5-diphenyltetrazolium bromide; Bimake, Houston, TX, USA) was performed to determine the number of viable cells in culture, as described by the manufacturer. MCF-7 and SKBR-3 cells were plated into 12-well plates at a density of 1 × 105 cells/well in 1 mL of DMEM. Cells were then treated with various concentrations of hydroalcoholic extraction of Vermentino leaves (100, 200, and 400 µg/mL) for 16 and 24 h. Cell proliferation rates were determined by adding 10 µL of MTT labeling reagent to each well and incubating at 37 °C for 30 min. The absorbance was measured using a plate reader to read the formation of formazan orange dye at 450 nm.

### 2.8. Flow Cytometry Assay

To determine the effects of Vermentino extract on cell apoptosis, MCF-7 and SKBR-3 cells were seeded at a density of 2 × 10^5^ cells per well in a 6-well plate and incubated at 37 °C overnight. The cells were treated with Vermentino extract for 16 or 24 h at concentrations of 100, 200, and 400 µg/mL and analyzed for apoptosis using flow cytometry. Untreated cells were used as a control (stimulated cells in the presence of the vehicle, water). Next, cells were trypsinized, washed with cooled PBS, and then resuspended in 200 µL binding buffer containing 5 µL annexin V-fluorescein isothiocyanate (FITC) and 10 µL propidium iodide (PI) (Beyotime Institute of Biotechnology, Jiangsu, China) to be incubated for 15 min in a dark room at room temperature, as suggested by the manufacturer. A fluorescence-activated cell sorter (FACS) Calibur analyzer (BD Biosciences, San Jose, CA, USA) was used to perform the analysis.

### 2.9. Clonogenicity Assay Analysis

Clonogenicity assays were performed to determine the cytotoxic effect of the hydroalcoholic extract of Vermentino on breast cancer cell lines. Briefly, MCF-7 and SKBR-3 cells were harvested from a stock culture, and cells were seeded into 12-well plates at a density of 250 cells per well 1 day before treatment started. The following day, the cells were treated with either the hydroalcoholic extract at concentrations of 100, 200, or 400 µg/mL and incubated for 16 or 24 h. After approximately 7–10 days of colony formation, colonies were fixed (paraformaldehyde 4% in PBS) and stained with crystal violet solution (Cell Biolab SNC, San Diego, CA, USA). The number of colonies were then counted. The cell-survival fraction was calculated by dividing the number of obtained colonies after treatment by the number of colonies in the control. Data were represented as number of colonies per well.

### 2.10. Proteomic Sample Preparation

Fifty micrograms of total proteins were extracted from control and 400 µg/mL Vermentino hydroalcoholic extract treated samples in SKBR-3 and MCF-7 cell lines. Proteins were denatured in a solution of 100 mmol/L Tris-HCl, pH 8, and 8 mol/L urea. Samples were then ultrasonicated for 10 min at intervals of 10 s on and 10 s off (QSonica, Newtown, CT, USA) and centrifuged at 16,000 × *g* for 10 min at 4 °C to remove insoluble pellets. The soluble fraction then was reduced with dithiothreitol (15 mM) for 1 h at room temperature and alkylated with iodoacetamide (15 mmol/L) for 30 min at room temperature in the dark. Then, 50 μg protein was diluted to a final concentration of 2 mol/L urea with 100 mmol/L Tris-HCl, pH 8, and digested overnight on a shaker at 37 °C in 3 μg of trypsin/Lys-C mix (Promega, Madison, WI, USA). Samples were de-salted and cleaned using hydrophobic-lipophilic balance (HLB) plates (Oasis, HLB 3 0 μm, 5 mg sorbent; Waters Corporation, Milford, MA, USA).

### 2.11. LC–MS/MS Settings

LC–MS/MS was performed on a Dionex Ultimate 3000 NanoLC connected to an Orbitrap Elite (Thermo Fisher, Waltham, MA, USA) equipped with an EasySpray ion source. The mobile phase A was composed of 0.1% aqueous formic acid, and mobile phase B was composed of 0.1% formic acid in acetonitrile. Peptides were loaded onto the analytical column (PepMap RSLC C18 2 μm, 100 Å, 50 μm i.d. × 15 cm) at a flow rate of 300 nL/min using a linear AB gradient composed of 2–25% A for 185 min, 25–90% B for 5 min, and then an isocratic hold at 90% for 5 min with re-equilibrating at 2% A for 10 min. The temperature was set to 40 °C for both columns. Nano-source capillary temperature was set to 275 °C, and spray voltage was set to 2 kV. MS1 scans were acquired in the Orbitrap Elite at a resolution of 60,000 full width at half maximum with an automated gain control target of 1 × 10^6^ ions over a maximum of 500 ms. MS2 spectra were acquired for the top 15 ions from each MS1 scan in normal scan mode in the ion trap with a target setting of 1 × 10^4^ ions, an accumulation time of 100 ms, and an isolation width of 2 Da. Normalized collision energy was set to 35%, and one microscan was acquired for each spectra.

### 2.12. Preparative Data Analysis and Peptide Identification Search

The raw MS files were converted to mzXML using MSConvert and searched against the Swiss-Prot reviewed human FASTA database using the COMET, X! Tandem native, and X! Tandem k-score search algorithms [20,21]. Precursor and fragment mass tolerance for all of these search algorithms were set to 10 ppm. Cysteine carbamidomethyl was set as a fixed modification, and oxidation of methionine was set as a variable modification for all search algorithms. A maximum of two missed cleavages for all tryptic peptides was allowed, and Target-decoy modeling of peptide spectral matches was performed with PeptideProphet [22]. Peptides with a probability score of >95% from the entire experimental dataset were imported into Skyline software [23,24] to establish a library for quantification of precursor extracted ion intensities (XICs). Precursor XICs from each experimental file were extracted against the Skyline library, and peptide XICs with isotope dot product scores >0.8 and with a minimum of two prototypic peptides per protein were filtered for final statistical analysis of proteomic differences [25]. Normalization of raw peptide intensities and protein level abundance inference was calculated using the linear mixed effects model built into the open sources MSSTATs (v3.2.2) software suite [26]. The mass spectrometry proteomics data were deposited to the ProteomeXchange Consortium via the PRIDE [27] partner repository with the dataset identifier PXD016748.

### 2.13. Protein Stability Assay and Half-Life Determination

Cycloheximide (60 μg/mL) was added to SKBR-3 cells in co-treatment with 400 µg/mL of Vermentino hydroalcoholic extract per well for 4, 18, and 24 h. Protein levels were determined at indicated time points by Western blotting, as described above, using the anti-FASN antibody. The relative amount of FASN protein was evaluated by densitometry and normalized to β-actin. Protein half-life was determined for each experiment using linear regression analysis.

### 2.14. Proteasomal Activity Assays

Proteasomal activity in MCF-7 and SKBR-3 cell lines was determined with luminescent site-specific substrates using the Proteasome-Glo assay (Promega), according to the manufacturer’s protocol. Briefly, cells were plated in 96-well plates at 5 × 10^3^ cells per well and incubated for 24 h. Cells were treated with Vermentino hydroalcoholic extract, at various concentrations (100, 200, and 400 µg/mL) for 16 and 24 h. Cells were assayed using the Proteasome-Glo kit for chymotrypsin-like (Chym-L), trypsin-like (Tr-L), and caspase-like (Casp-L) activity, according to the manufacturer’s protocol.

### 2.15. RNA Interference

To perform the RNAi experiments, six different predesigned small interfering RNAs (siRNAs) targeting human SUMO1 (#1 sense sequence: 5’-GGAUAGCAGUGAGAUUCACtt-3’, antisense: 3’-GUGAAUCUCACUGCUAUCCtc-5’), (#2 sense sequence 5’-AAGGUGAAUAUAUUAAACUCA-3’ and antisense: 3’-UGAGUUUAAUAUAUUCACCUU-5’), human SUMO2 (#1 sense sequence: 5’-CCACAUCCUGACUACUACCtt -3’, antisense: 3’-GGUAGUAGUCAGGAUGUGGtg-5’), (#2 sense sequence: 5’-GCUGUUACAUGUAGGGCAATT-3’, antisense: 3’-UUGCCCUACAUGUAACAGCTA -5’), and human SUMO3 (#1 sense sequence: 5’-GGCAGAUCAGAUUCAGGUUtt -3’, antisense: 3’-AACCUGAAUCUGAUCUGCCtc -5’) were purchased from Ambion (Austin, TX, USA). The siRNA sequence used for silencing of SUMO3 #2 corresponds to the coding region 161–179 (relative to the start codon), as previously described [28] (Dharmacon Research, Inc., Boulder, CO, USA). MCF-7 and SKBR-3 cells were cultured in 6-well plate (0.5 × 10^6^ cells/well) and transfected using RNAiMax (5 μL/well) (Invitrogen, Carlsbad, CA, USA) with SUMO1 siRNA (10 nM), SUMO2 siRNA (10 nM), or SUMO3 siRNA (10 nM) for 48 h for mRNA or protein expression analyses.

### 2.16. Statistical Analysis

Data are expressed as the mean ± standard error of the mean (SEM) or mean ± standard deviation (SD). Statistical analysis was performed using analysis of variance and Fisher’s exact test and *t*-test. Significance was defined by *p* < 0.05.

## 3. Results

### 3.1. Polyphenol Phytochemical Composition

To determine the phenolic profile of Vermentino, we subjected crude leaf extract to HPLC chromatography. We found that Vermentino leaves contain a mixture of bioactive compounds, which primarily include quercetin 3-O glucoside and isorhamnetin glucoside (Table 1).

### 3.2. Vermentino Extract Lowers Cell Viability in Breast Cancer Cell Lines

As phenolic compounds, such as flavonoids, have been shown to induce cytotoxicity in cancer cell lines, we evaluated cell viability after treatment with Vermentino hydroalcoholic extract. We treated MCF-7 and SKBR-3 breast cancer cell lines as well as the MCF-12A human breast epithelial cell line and compared the resulting cytotoxicity. All three cell lines were treated with Vermentino hydroalcoholic extract in increasing concentrations of 100, 200, and 400 µg/mL for 16 and 24 h. Vermentino compound showed significant cytotoxic activity at each concentration, with a 50% reduction of viability following treatment of MCF-7 and SKBR-3 cell lines with 400 µg/mL of the hydroalcoholic extract compared to the control (Figure 1A). Importantly, MCF-12A cells showed no reduction in viability over time with Vermentino hydroalcoholic extract. Thus, Vermentino hydroalcoholic extract showed slight apoptotic effects on MCF-7 and SKBR-3 breast cancer cells but not on MCF-12A epithelial breast cells.

### 3.3. Vermentino Extract inhibits Clonogenicity in Breast Cancer Cell Lines

To assess the survival and proliferation of MCF-7 and SKBR-3 human breast cancer cells treated with Vermentino hydroalcoholic leaf extract, we performed clonogenicity assays. Colony formation was evaluated with crystal violet stain, and representative colonies are shown in Figure 1B,C. Vermentino compound was used in concentrations of 100, 200, and 400 µg/mL for 16 and 24 h. The pure hydroalcoholic dilutions of 200 and 400 µg/mL at 24 h were highly toxic and resulted in 48% and 80% reductions in colony formation in MCF-7 cells, respectively, compared to control colonies (Figure 1B). Additionally, the ethanolic dilutions of 200 and 400 µg/mL showed 39.1% and 78.3% reductions in colony formation in SKBR-3 cells, respectively, at 24 h compared to control colonies (Figure 1C). These results suggest that the Vermentino compound is a promising promoter of cytotoxicity in the breast cancer cells tested.

### 3.4. Vermentino Extract Induced Late Apoptosis/Necrosis in Breast Cancer Cell Lines

To examine whether the cytotoxic activity of Vermentino leaf hydroalcoholic extract was due to the induction of apoptosis, cells were exposed to 100, 200, and 400 µg/mL hydroalcoholic extract for 16 and 24 h and stained with annexin V-FITC and propidium iodide (PI) and were then analyzed by cytofluorometry. Treatment for 16–24 h with 200 and 400 μg/mL Vermentino leaf hydroalcoholic extract triggered a significant increase in early apoptosis in 20% of MCF-7 and SKBR-3 cells compared to the control (Figure 2A,B). Additionally, the percentage of cell necrosis was significantly increased compared to the control after exposure to 100 μg/mL of the leaf hydroalcoholic extract for 16 and 24 h in MCF-7 and SKBR-3 cells (Figure 3A,B). Collectively, incubation with Vermentino leaf hydroalcoholic extract for 24 h significantly increased late apoptotic cells by 20% and necrosis by 25% compared to the control in both cell lines, with the greatest effect observed at a dosage of 400 μg/mL These results indicate that treatment with Vermentino extract increases cell death via late apoptosis and necrosis in both breast cancer cell lines.

To assess the alterations occurring after treatment with Vermentino extract, we performed an unbiased proteomic screen by LC–MS/MS analysis to investigate which cell signaling pathways were involved causing cytotoxicity. According to the proteomic results, most of the apoptosis pathways were deregulated in breast cancer cell lines after treatment with Vermentino compound (Appendix A; Appendix A).

### 3.5. Vermentino Extract Induced CASP-9 and CASP-3 in Breast Cancer Cell Lines

To further investigate the apoptotic potential of Vermentino extract on MCF-7 and SKBR-3 breast cancer cell lines, we assessed the expression and cleavage of caspase-9 (CASP-9) and caspase-3 (CASP-3), which are closely associated with cell death via apoptosis. As shown in Figure 3A, pro-caspase-9 protein levels were reduced to 32% and 45% at concentrations of 200 and 400 µg/mL Vermentino hydroalcoholic extract, respectively, after 24 h in MCF-7 cells compared to the control. Consequently, cleaved caspase-9 expression increased by 1.2 at a concentration of 400 µg/mL hydroalcoholic extract compared to the control. Cleavage of caspase-9 implied that apoptosis was induced by Vermentino extract. Furthermore, pro-caspase-3 activation was induced by exposure to the phytochemical hydroalcoholic extract of Vermentino, resulting in 25% and 45% reductions of pro-caspase-3 compared to the control at concentrations of 200 and 400 µg/mL, respectively, after 24 h. Consistent with activation of the caspase cascade, cleaved caspase-3 was markedly increased by 2.23- and 2.2-fold at a concentration of 400 µg/mL Vermentino hydroalcoholic extract, respectively, after 24 h.

Consistent with our observations in the MCF-7 cell line, Vermentino extract had a strong activation effect on the caspase cascade in SKBR-3 cells. These results indicate that Vermentino hydroalcoholic extract strongly activated pro-caspase-9, which showed a decrease of 35% at a concentration of 200 µg/mL and a decrease of 56% at a concentration of 400 µg/mL after 24 h compared to the control. Cleaved caspase-9 showed increases of 2.4-fold at a concentration of 400 µg/mL Vermentino hydroalcoholic extract compared to the control. Pro-caspase-3 activation was induced by exposure to the hydroalcoholic extract of Vermentino, as evidenced by pro-caspase-3 reductions of 20% and 60% at concentrations of 200 and 400 µg/mL, respectively, after 24 h compared to the control. Cleaved caspase-3 expression increased, implying that cell apoptosis was induced by Vermentino phytochemical compound.

### 3.6. Vermentino Extract Lowers FASN Protein Level and Promoted AKT1 Signaling in Breast Cancer Cell Lines

Our proteomic analysis revealed that FASN is one of the key apoptosis-related proteins that resulted in being downregulated by Vermentino extract (Appendix A). Figure 3B and Appendix A show the intensity of the extracted precursors isotopic envelope (M, M + 1, M + 2) of a representative FASN peptide in control and Vermentino hydroalcoholic extract in MCF-7 and SKBR-3 after 24 h of 400 µg/mL treatment. MCF-7 and SKBR-3 cells treated with Vermentino hydroalcoholic extract showed 70% and 20% reductions in FASN peptide intensity, respectively.

To further confirm the effect of Vermentino leaf extract on cell apoptosis, we treated MCF-7 and SKBR-3 cell lines with 100, 200, and 400 µg/mL hydroalcoholic extract for 24 h and examined the protein expression of FASN, AKT1, and p-AKT1. As shown in MCF-7 cells in Figure 3C, FASN protein levels decreased by up to 37% at the protein level at a concentration of 400 µg/mL, whereas AKT1 levels remained relatively constant compared to the control. In contrast, p-AKT1 showed an increase of 2.6- and 4.17-fold at 200 and 400 µg/mL, respectively, compared to the control. However, FASN mRNA levels showed a slight increase of 1.2- and 1.6-fold at 200 and 400 µg/mL, respectively, compared to the control (Figure 3C, left panel). Similar effects were observed in SKBR-3 cells (Appendix A). FASN protein levels decreased by 9% and 35% at concentrations of 200 and 400 µg/mL, respectively, whereas p-AKT1 increased by 3.6- and 4.8-fold at the same respective concentrations compared to the control. Consistent with the data described above, SKBR-3 cells showed little variation in protein levels of AKT1 compared to the control. However, FASN mRNA levels rose significantly in both cell lines (Appendix A, left panel). Together, these results indicate that Vermentino hydroalcoholic extract affects FASN at the level of translation rather than transcription.

In addition, proteomic analysis revealed TRAP1 (tumor necrosis factor (TNF) receptor associated protein 1) significantly downregulated after treatment with Vermentino extract (Appendix A). TRAP1 is the mitochondrial member of the heat shock protein 90 (HSP90) family that directly interacts with respiratory complexes, regulates mitochondrial permeability transition in response to apoptotic stimuli, and mediates mitochondrial death [29].

### 3.7. Vermentino Extract Lowered FASN Protein Expression by Activating Proteasomal Degradation

On the basis of the observation that FASN was regulated at the post-transcriptional level and was reduced by treatment with Vermentino extract, we next examined the protein synthesis and degradation of FASN protein. First, we treated MCF-7 and SKBR-3 cells with 400 µg/mL Vermentino hydroalcoholic extract for increasing amounts of time. We found that FASN levels were progressively reduced over time. As shown in MCF-7 cells in Figure 4A, FASN protein levels fell by 16% and 26% after 18 and 24 h of exposure, respectively, compared to the control. Similarly, FASN protein levels decreased by 29% and 60% in SKBR-3 cells after 18 and 24 h of compound exposure, respectively, compared to the control (Figure 4A, right panel). We went on to calculate FASN protein half-life in MCF-7 cells, which decreased from 42.8 ± 7 h in cycloheximide (CHX)-treated samples to 23.7 ± 3 h in CHX and Vermentino hydroalcoholic extract co-treated samples, suggesting that FASN synthesis was not affected by Vermentino extract (Figure 4B, left panel). Interestingly, loss of FASN protein was significantly reduced by pretreatment with the proteasome inhibitor MG132 (Figure 4B, right panel), indicating that protein degradation may have been impacted.

We next investigated whether FASN protein degradation was dependent on ubiquitination status. Figure 4C shows that treatment with 400 µg/mL Vermentino hydroalcoholic extract for 24 h increased FASN ubiquitination by threefold compared to the control. In addition, treatment with MG132 prevented Vermentino-mediated degradation of FASN protein levels in MCF-7 and SKBR-3 cell lines and resulted in intracellular accumulation of ubiquitinated FASN.

### 3.8. Effects of Vermentino Extract on Proteasomal Activity

Because MG132 protected against Vermentino-induced FASN degradation, we next investigated whether proteasomal activity was altered by treatment with the Vermentino compounds by measuring trypsin-like (Tr-L), caspase-like (Casp-L), and chymotrypsin-like (Chym-L) proteasomal activity. Treatment with hydroalcoholic extract for 16 and 24 h significantly decreased Tr-L activity in MCF-7 (Appendix A) and SKBR-3 (Appendix A) cell lines in a dose-dependent manner. After 16 and 24 h of treatment, Tr-L activity was reduced most significantly when treated with 400 µg/mL of each compound in both cell lines. Neither of the Vermentino extracts had an effect on Casp-L activity in MCF-7 (Appendix A) nor SKBR-3 (Appendix A) cell lines. Although Chym-L activity was decreased in MCF-7 and SKBR-3 cell lines compared to controls by 50%, there was no significant difference across doses or times (Appendix A, respectively).

### 3.9. Vermentino Hydroalcoholic Extract Lowered UBC9 Protein Level in Human Breast Cancer Cell Lines

As no alterations in Casp-L proteasomal activity were observed with either Vermentino leaf extract, we began investigating SUMOylation, a post-translational modification that causes proteasomal degradation in crosstalk with ubiquitination [30]. Our proteomic data showed that the sole E2 enzyme driving SUMOylation, ubiquitin-conjugating enzyme E2 I (UBC9), was downregulated by Vermentino treatment in both cell lines (Appendix A). Interestingly, UBC9 mediates the stability of its target proteins and is upregulated in premalignant conditions [30,31]. Therefore, we evaluated the protein level of UBC9 and found that UBC9 decreased by 80% and 50% in response to 24 h of exposure to 400 µg/mL Vermentino hydroalcoholic extract in MCF-7 and SKBR-3 cell lines, respectively (Figure 5). The intensity of the extracted precursor isotopic envelope (M, M + 1, M + 2) of a representative UBC9 peptide is shown in MCF-7 cells (Figure 5, top left panel) and SKBR-3 cells (Figure 5, bottom left panel) after treatment with 400 µg/mL Vermentino hydroalcoholic extract for 24 h. MCF-7 and SKBR-3 Vermentino-treated cell lines both showed a 40% reduction in UBC9 M + 2 peptide intensity.

### 3.10. SUMOylation Mediated FASN Protein Stability

Modulation of protein SUMOylation or deSUMOylation modification has been associated with regulation of carcinogenesis in breast cancer [32]. Using three different prediction software, we found that FASN was a highly probable target for SUMOylation (Table 2), thus this led us to examine the in vivo and in vitro SUMOylation of FASN. We found that human breast cancer tissues exhibited high FASN SUMOylation compared to normal breast tissue (Figure 6A). Individual silencing of SUMO1, SUMO2, and SUMO3 in MCF-7 cells did not affect FASN mRNA levels, suggesting that SUMOylation affects FASN protein stability at the post-translational level (Figure 6B). Western blot analysis further implicated SUMO2 in downregulation of FASN protein level. Specifically, silencing SUMO2 resulted in a 70% drop in FASN protein level compared to control in MCF-7 cells (Figure 6C). Figure 6D shows that treatment of MCF-7 cells with Vermentino hydroalcoholic extract for 24 h at 400 µg/mL inhibited formation of the SUMOylation complex by lowering UBC9 protein level without affecting SUMO2 protein expression. To provide evidence that FASN requires SUMO2 for stabilization and protection against degradation by the proteasome, MCF-7 cells were treated with CHX for 18 and 24 h in combination with SUMO2 silencing (Figure 6E, left panel). Co-treated cells exhibited reductions in FASN protein levels of 70% and 80%, respectively, compared to the control. In contrast, MCF-7 cells treated with MG132 for 18 and 24 h in combination with SUMO2 silencing did not exhibit a significant drop in FASN protein levels (Figure 6E, right panel). Together, these results suggest that SUMO2 is indeed required for FASN stabilization.

## 4. Discussion

Many studies have demonstrated positive associations between cytotoxicity in cancer cells and antioxidant activities of plant-derived compounds, such as quercetin, acid gallic, and isorhamnetin [33,34]. In this study, we evaluated the potential cytotoxic effects of the hydroalcoholic extract of Vermentino leaves.

The phenolic profile obtained by HPLC-UV analysis suggests that the Vermentino leaf extract may have cytotoxic capability. Indeed, we found that treatment with the extract suppressed cell viability and survival in MCF-7 and SKBR-3 breast cancer cell lines and exerted no significant effect on MCF-12A normal breast epithelial cells. Moreover, cytofluorometric analyses suggested that breast cancer cells were driven toward late apoptosis and necrosis in a dose- and time-dependent manner. The involvement of the apoptotic pathway was further verified by IPA software [35], which showed that the most affected canonical biological pathway was apoptosis.

To investigate the potential pathways underlying the cytotoxic activity, we performed proteomic analyses on treated cells and determined that Vermentino leaf extract downregulated fatty-acid synthase (FASN) expression at the protein level. FASN is a key enzyme in fatty-acid biosynthesis and plays an important role in energy homeostasis via β-oxidation [36]. Lipids are the main component of the cell membrane and are essential for cell division. Tumor cells endogenously synthesize extra fatty acids via de novo fatty-acid synthesis, which allows them to sustain higher proliferation rates and faster growth [37]. Inhibition of FASN induces apoptosis and creates cytotoxicity that is likely to trigger cell death due to the accumulation of malonyl-CoA [38,39]. To evaluate the mechanism underlying induction of apoptosis in MCF-7 and SKBR-3 cells exposed to Vermentino leaf extract, we evaluated caspase-3 and caspase-9 activity. We found that the Vermentino leaf extract induced programmed cell death in the breast cancer cells via modulation of the caspase-dependent pathway. Consistent with our results, elevation of cleaved caspases is a crucial molecular target in chemoprevention [40,41]. The involvement of caspase activation in FASN-induced apoptosis was evaluated by Western blot. We found that cleaved caspases increased in a dose-dependent manner after 16- and 24-h treatments with Vermentino leaf extract.

FASN expression is regulated by several growth factors, including epidermal growth factor receptor (EGFR), receptor tyrosine-kinase (ERBB2), and steroid hormone receptors (estrogen receptor, progesterone receptor, and androgen receptor) [42]. Binding of growth factors and receptors results in activation of the phosphoinositide 3-kinase (PI3K)-AKT transduction pathway. FASN activity and AKT1 activation are modulated in a coregulatory pathway—FASN regulates AKT1 and activated AKT1 regulates FASN. AKT1 activation protects cells against FASN inhibition-induced cell death by increasing transcription of FASN [43,44]. Our data show that Vermentino leaf extract triggered activation of AKT1 via phosphorylation as well as increased FASN mRNA levels in MCF-7 and SKBR-3 cells. Activated AKT1 stimulates FASN expression by gene modulation of sterol regulatory element-binding protein 1c (SREBP1C), which is a transcription factor that activates FASN by binding to its promoter region [45]. Here, we provide evidence that increased AKT1 activation and the related increase in FASN mRNA compensate for Vermentino-induced FASN downregulation.

To further understand the mechanism of FASN reduction, we explored two pathways: protein synthesis and proteasomal degradation. Treatment with cycloheximide (CHX), a known protein synthesis inhibitor, decreased FASN protein expression by locking protein turnover. However, co-treatment with CHX and Vermentino leaf extract enhanced the reduction of FASN compared to CHX alone, suggesting that downregulation of FASN by the compound is independent of protein synthesis. In contrast, pretreatment with the proteasomal degradation inhibitor MG132 protected FASN from degradation during co-treatment with Vermentino leaf extract, suggesting that the extract downregulates FASN accumulation via activation of proteasomal degradation. Indeed, previous work has demonstrated that treatment with chemotherapeutic agents lowers proteasomal activity of the 20s subunit [46].

Consistent with our data, we found a downregulation of Tr-L and Chym-L activity and no change in Casp-L activity in both cell lines, compared to controls. These results suggest that FASN downregulation is mediated by Casp-L activity, as Casp-L activity was not affected by Vermentino-induced downregulation. FASN inhibitors cause the accumulation of ubiquitinated proteins through an undetermined mechanism [47]. Similarly, Vermentino leaf extract increased the ubiquitination of FASN. Although we observed a general reduction in proteasomal activity, affinity for the proteasomal 20s subunit was increased and FASN degradation was promoted by proteasomal Casp-L activity.

Together, these data suggest a level of crosstalk between the fatty-acid synthesis and proteasomal pathways whereby FASN reduction is mediated by increased ubiquitination. Therefore, Vermentino leaf extract act to modulate FASN expression and degradation via two distinct mechanisms. The first mechanism involves AKT1 activation, which induces transcription of FASN mRNA. The second mechanism promotes FASN degradation via the ubiquitin-proteasome pathway. Our findings suggest that the homeostatic balance in breast cancer cells favors the second pathway and allows Vermentino-induced FASN degradation to outweigh the induction of FASN mRNA transcription.

Cancer cells sustain high proliferation levels by relying on de novo fatty-acid synthesis in addition to glycolysis. Here, we demonstrated that treatment of breast cancer cells with Vermentino leaf extract altered the levels of FASN and disrupted energetic metabolism, resulting in apoptosis. Additionally, TRAP1 (TNF receptor associated protein 1), a molecular chaperone involved in the regulation of energetic metabolism in several cancers [29] including breast cancer, was significantly downregulated after treatment with Vermentino extract, in accordance with the proteomic results [28,48]. This disruption in the Warburg phenotype [49] of cancer cells leads to increased oxygen consumption rates, which cannot be sustained and ultimately lead to cell death. Synergistic effects of Vermentino-induced FASN and TRAP1 downregulation may drive breast cancer cells to apoptotic cell death and exhibited no effect on normal cells.

Our proteomic analysis also revealed strong reduction in UBC9, the sole E2 enzyme that drives SUMOylation. SUMOylation is a small ubiquitin (UB)-related modifier (SUMO) conjugation that is similar to ubiquitination [30,31]. However, unlike ubiquitination, which normally targets proteins for degradation, SUMOylation regulates protein stability. UBC9 is the only well-characterized E2 enzyme in the SUMOylation cycle that transfers activated SUMO to the target protein [30]. UBC9 and SUMO are highly expressed in human premalignant conditions in response to low-grade, long-term oxidative stress, suggesting that up-regulation of SUMOylation may be an adaptive response to oxidative stress [31,50]. Therefore, antioxidant treatments could be expected to reduce SUMOylation levels. On the basis of our proteomic results, we hypothesized that FASN may be a target for SUMOylation, contributing to protein stability; in addition, it was highly predicted to be SUMOylated. Immunoprecipitation of FASN in breast cancer cells treated with and without Vermentino extract confirmed our hypothesis that UBC9 downregulation results in reduced SUMOylation complex formation. To understand which SUMO protein was interacting with FASN, we silenced SUMO1, SUMO2, and SUMO3 and found that only SUMO2 silencing reduced the FASN protein level without affecting individual SUMO expression. This suggests that UBC9 drives SUMOylation of FASN through SUMO2 and that SUMO2 acts in a protective mechanism to prevent FASN from being targeted for degradation. Silencing SUMO had no effect on transcription of FASN mRNA, demonstrating that FASN downregulation occurs at the post-translational level. To demonstrate that SUMO2 protects FASN against proteasomal degradation, we treated the breast cancer cell lines with a protein synthesis inhibitor (CHX) and a proteasomal degradation inhibitor (MG132) in combination with SUMO2 silencing. We found that, in combination with CHX treatment, SUMO2 silencing reduced FASN protein levels more than SUMO2 silencing alone. Inhibiting proteasomal degradation had no effect on FASN protein expression.

## 5. Conclusions

In summary, we demonstrated for the first time not only that FASN is SUMOylated, but also that Vermentino leaf extract reduces the ability of breast cancer cells to SUMOylate FASN by lowering the UBC9 protein level. These results provide evidence for novel therapeutic capabilities of Vermentino leaf extract in breast cancer.

## Figures and Tables

**Figure 1 biomolecules-10-00529-f001:**
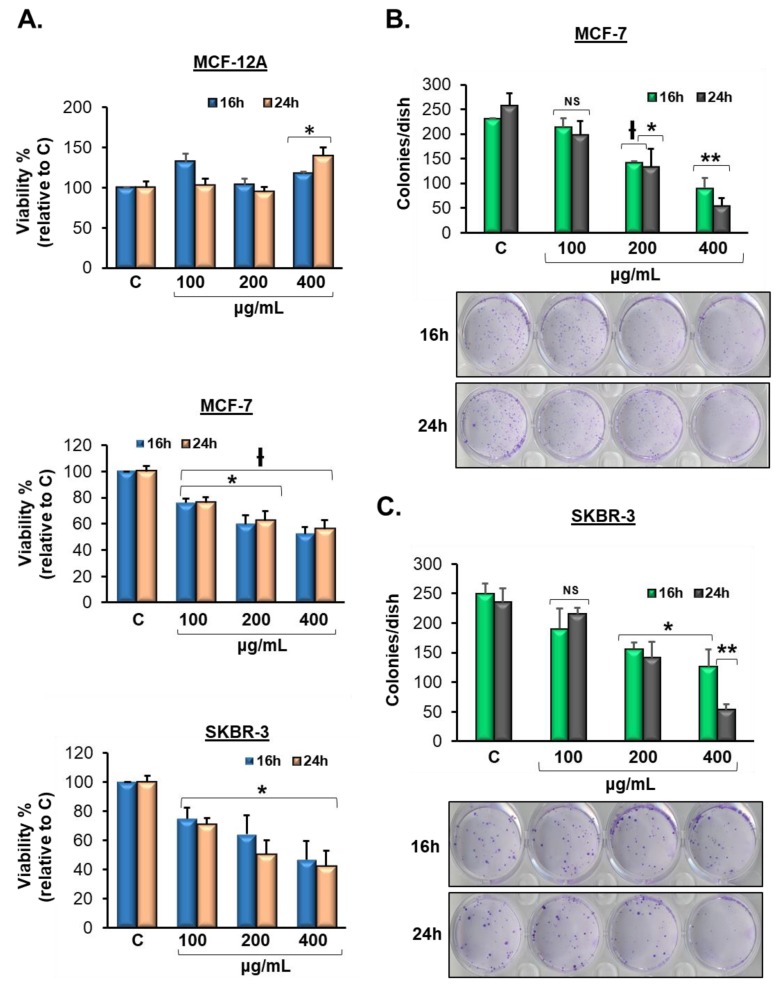
Vermentino extract lowers breast cancer cells’ viability and clonogenicity in a dose-specific response. (**A**) MTT (3-(4,5-dimethylthiazol-2-yl)-2,5-diphenyltetrazolium bromide) assay showing MCF-12A, SKBR-3, and MCF-7 cells treated with hydroalcoholic extract of Vermentino (100, 200, or 400 µg/mL) for 16 and 24 h. The absorbance of the formazan dye was measured at 450 nm. The results are expressed as percentage of cell growth (%) ± SE from three independent experiments performed in triplicate. * *p* < 0.05 vs. control. ^Ɨ^
*p* < 0.005 vs. control. (**B**) MCF-7 and (**C**) SKBR-3, 0.1 × 10^6^ cells (6-well plates) were plated to perform clonogenicity assay adding Vermentino hydroalcoholic extract for 16 and 24 h at concentrations of 100, 200, and 400 µg/mL. Photographs of the Petri dishes in a representative experiment are shown below the histograms. All data are expressed as colonies per dish in three independent experiments performed in triplicate. * *p* < 0.05, ** *p* < 0.001, ^Ɨ^
*p* < 0.0001 vs. the control group.

**Figure 2 biomolecules-10-00529-f002:**
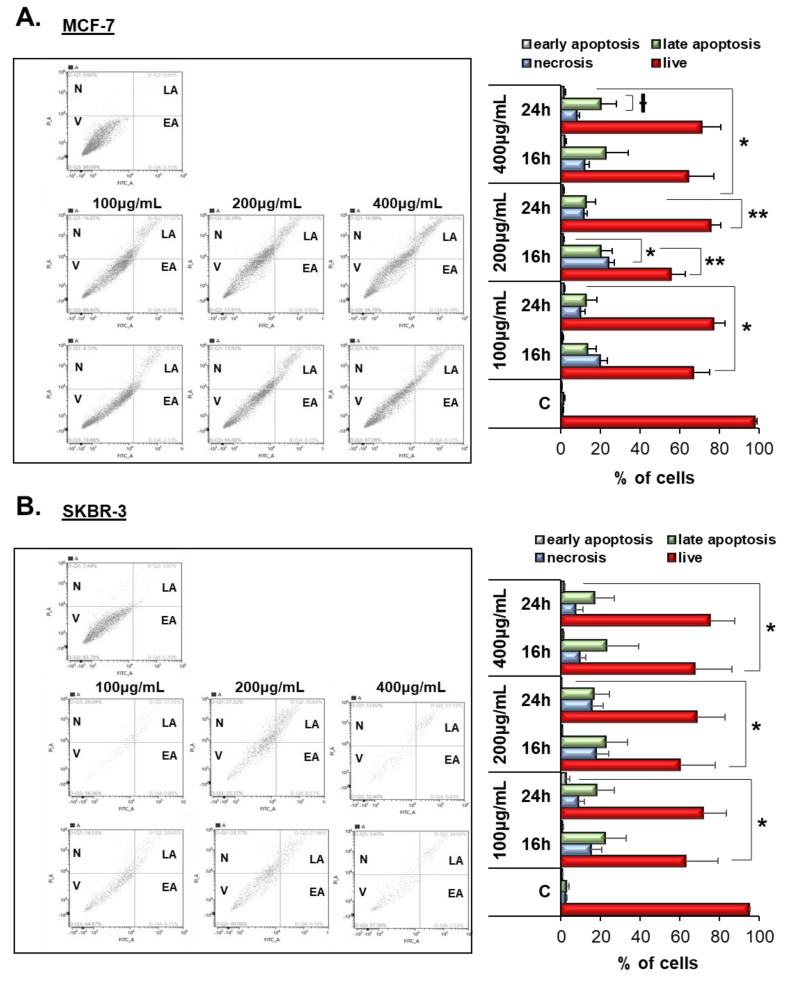
Hydroalcoholic Vermentino extract induces cell death via late apoptosis and necrosis. Apoptotic cell death of MCF-7 (**A**) and SKBR-3 (**B**) cells after treatment with Vermentino leaf hydroalcoholic extract after 16 and 24 h at concentrations of 100, 200, and 400 µg/mL. Apoptotic cells were measured using annexin V-FITC and propidium iodide staining assay. The lower left quadrant indicates viable cells (V), the upper left quadrant indicates necrosis (N), the upper right quadrant shows late apoptotic cells (LA), and the lower right shows the early apoptotic cells (EA). In the histogram, the values are reported as mean ± SE of three independent experiments performed in triplicate. * *p* < 0.05, ** *p* < 0.001, ^Ɨ^
*p* < 0.004 vs. the control group.

**Figure 3 biomolecules-10-00529-f003:**
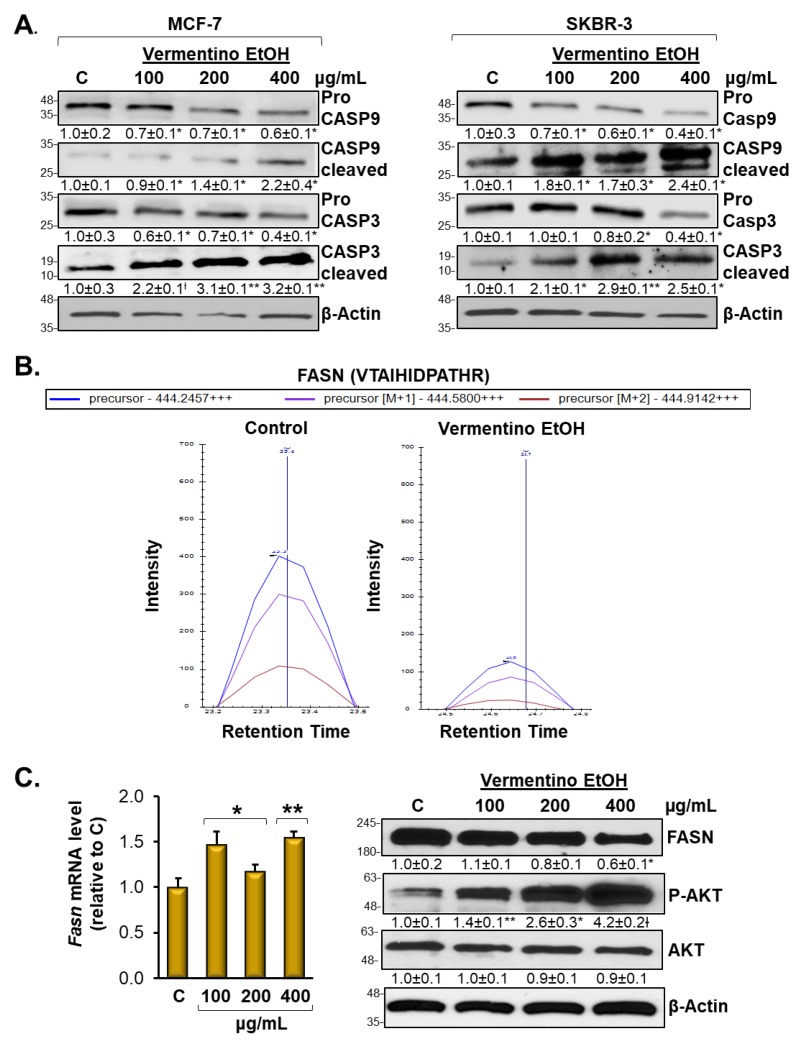
Vermentino extract induced caspase-3 (CASP-3) and caspase-9 (CASP-9) activation and lowered fatty-acid synthase (FASN) protein level. MCF-7 and SKBR-3 cells were plated at a density of 0.4 × 10^6^ cells in 6-well plates and treated with various concentrations of hydroalcoholic extract of Vermentino leaves (100, 200, and 400 µg/mL) for 24 h. (**A**) Western blot analysis of procaspase-9, cleaved caspase-9, procaspase-3, and cleaved caspase-3 expression in hydroalcoholic extract-treated MCF-7 and SKBR-3 cells. Densitometric ratios normalized to actin are shown below each Western blot. Results are expressed as fold of control (mean ± SE) from five independent experiments. * *p* < 0.05, ** *p* < 0.005, ^Ɨ^
*p* < 0.001 vs. control. (**B**) Intensity of the extracted precursor isotopic envelope (M, M + 1, M + 2) of a representative FASN peptide VTAIHIDPATHR (MCF-7 cells). All matched the theoretical isotopic distribution. (**C**) mRNA level of FASN was accomplished by using RT-PCR, whereas protein level was analyzed by Western blot analysis for FASN, AKT, and p-AKT (MCF-7 cells). Densitometric ratios normalized to β-actin are shown below the Western blot. Results are expressed as a fold of control (mean ± SE) from four and three independent experiments performed in triplicate for mRNA and protein analysis, respectively. * *p* < 0.05, ** *p* < 0.002, ^Ɨ^
*p* < 0.001 vs. control.

**Figure 4 biomolecules-10-00529-f004:**
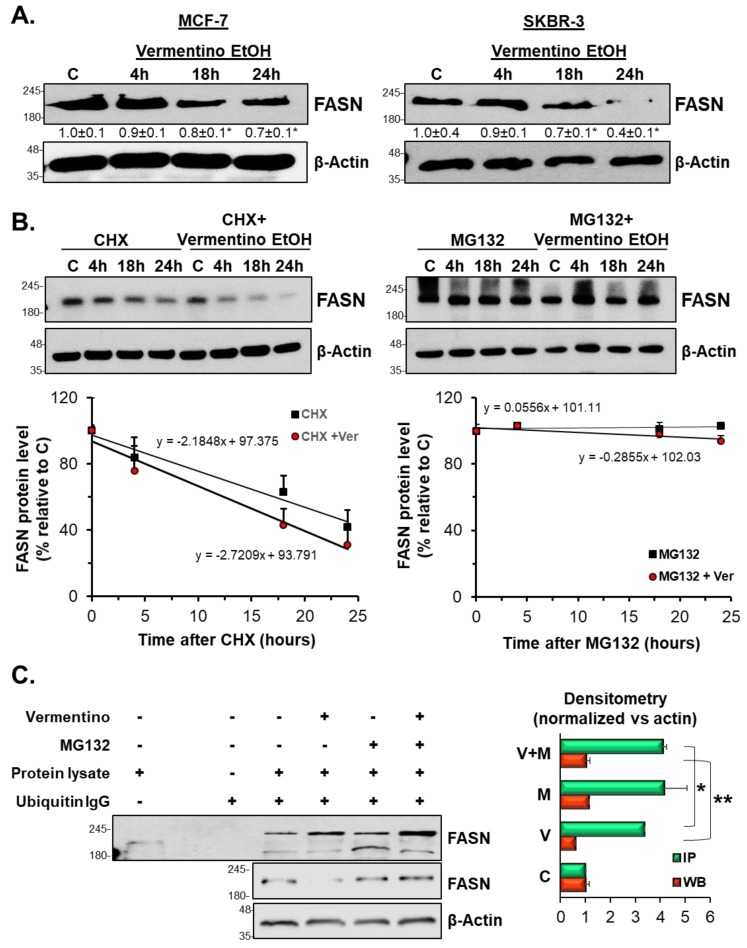
Vermentino hydroalcoholic extract lowers protein stability of FASN. (**A**) MCF-7 and SKBR-3 cells were treated in time course with 400 µg/mL Vermentino hydroalcoholic extract. Western blotting results are expressed as fold relative to control from three independent experiments and normalized with β-actin. * *p* < 0.05. (**B**) FASN protein stability was determined in MCF-7 cells by cycloheximide (CHX) and MG132 treatments when indicated, as described in the Methods section. Linear regression equation was used to calculate half-life. Results represent mean ± SEM from three independent experiments expressed as percentage of respective 0 h level. * *p* < 0.05, ** *p* < 0.008, N.C. 0.71 ≤ *p* ≤ 0.91. (**C**) Ubiquitination of FASN was analyzed by immunoprecipitation. Densitometric ratios normalized to β-actin are shown next to the Western blot. Results are expressed as a fold of control (mean ± SE) from three independent experiments performed in triplicate. * *p* < 0.005 vs. control (WB). ** *p* < 0.01 vs. control (IP).

**Figure 5 biomolecules-10-00529-f005:**
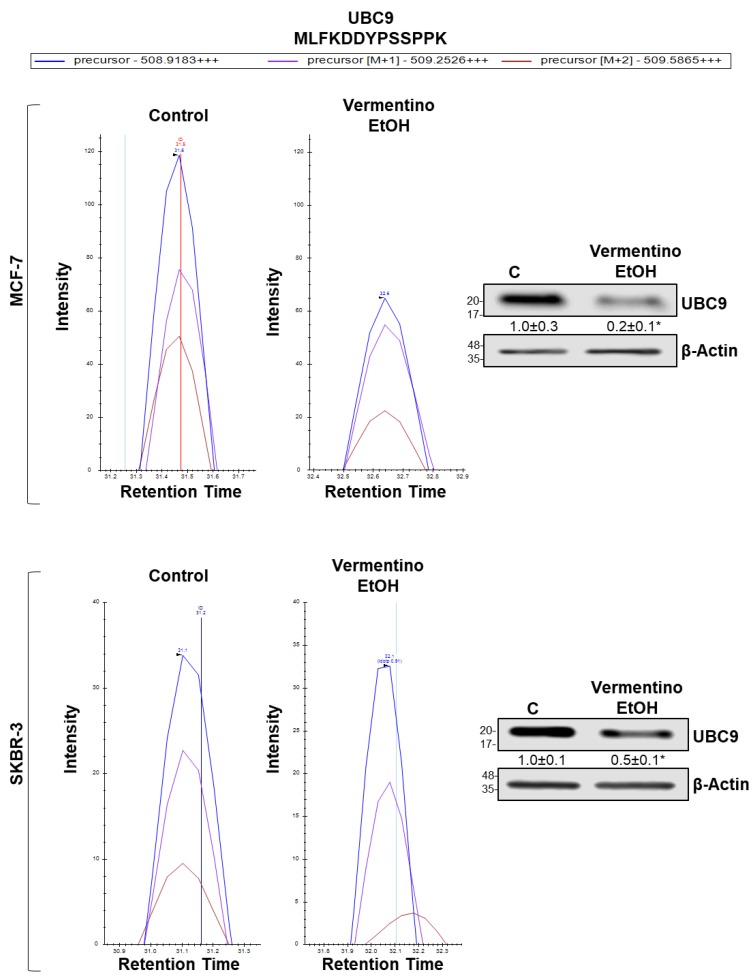
Vermentino hydroalcoholic extract reduces ubiquitin-conjugating enzyme 9 (UBC9) protein level in MCF-7 and SKBR-3 cells. Cells were treated with 400 µg/mL of extract for 24 h. Total proteins were extracted and analyzed by mass spectrometry and Western blotting UBC9 protein level. (Top and bottom left panels) Intensity of the extracted precursor isotopic envelope (M, M + 1, M + 2) of a representative UBC9 peptide MLFKDDYPSSPPK in MCF-7 and SKBR-3 cells treated with Vermentino hydroalcoholic extract. All matched the theoretical isotopic distribution. (Top and bottom right panels) Western blotting of UBC9. Densitometric ratios normalized to actin are shown below each Western blot. Results are expressed as fold of control (mean ± SE) from five independent experiments. * *p* < 0.03 (MCF-7 cells), ** *p* < 0.05 (SKBR-3 cells) vs. control.

**Figure 6 biomolecules-10-00529-f006:**
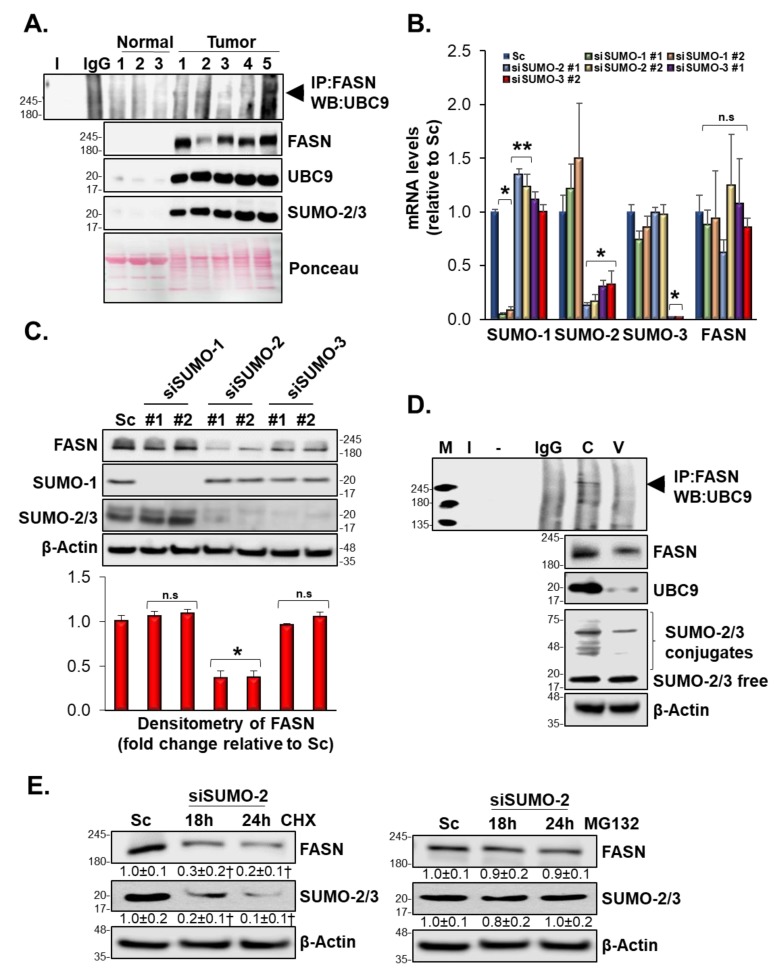
FASN was SUMOylated in vitro and in vivo and Vermentino hydroalcoholic extract inhibits the SUMOylation machinery in MCF-7 cells. (**A**) FASN immunoprecipitation in 700 µg of proteins from human breast normal (*n* = 3) and cancer tissues (*n* = 5) was blotted against UBC9. (**B**,**C**) Cells were transfected with siSUMO-1, siSUMO-2, and siSUMO-3 for 48 h. SUMO and FASN expression levels were analyzed by RT-PCR and Western blotting. (**D**) MCF-7 cells were treated with 400 µg/mL of extract for 24 h and FASN was immunoprecipitated to analyze the complex formation with UBC9. (**E**) Silencing of SUMO-2 was performed (48 h) in co-treatment with CHX and MG132 for the last 18 and 24 h. FASN and SUMO-2 protein levels were analyzed by Western blotting. Densitometric ratios normalized to actin and results are expressed as fold of control (mean ± SE) from three (**B**,**E**) and four (**C**) independent experiments. * *p* < 0.01, ** *p* < 0.04, *** *p* < 0.0003, † *p* < 0.002 vs. Sc.

**Table 1 biomolecules-10-00529-t001:** Quantification (g/kg ± SD) of phenols in the hydroalcoholic extract of Vermentino grapevine leaves, obtained by HPLC-UV analysis.

Compound	Vermentino (EtOH/H20)
Myricetin 3-O glucoside	0.11 ± 0.01
Eridictyol 7-glucoside	0.25 ± 0.03
Quercetin 3-O rutinoside	0.20 ± 0.02
Quercetin 3-O galactoside	1.29 ± 0.06
Quercetin 3-O glucoside	5.92 ± 0.10
Kaempferol 3-O galactoside	0.68 ± 0.03
Kaempferol 3-O rutinoside	0.08 ± 0.01
Kaempferol 3-O glucoside	1.64 ± 0.07
Quercetin 3-O-(6 acetyl) glucoside	0.19 ± 0.03
Isorhamnetin glucoside	8.00 ± 0.22
Total	18.36

**Table 2 biomolecules-10-00529-t002:** Predicted FASN small ubiquitin-like modifier (SUMO)ylation sites.

Position	Sequence	JASSA	SUMOsp	SUMOgo
K706	QELKKVIREPKPRSARWLSTS	X		X
K786	KPSCTIIPLMKKDHRDNLEFF	X	X	X
K835	RGTPLISPLIKWDHSLAWDVP	X	X	X
K1523	AFRHFLLEEDKPEEPTAHAFV	X	X	X
K1752	DLVLNSLAEEKLQASVRCLAT	X		X
K2206	EASELACPTPKEDGLAQQQTQ	X	X	X
K2471	DYNLSQVCDGKVSVHVIEGDH	X		X

## Data Availability

Proteomics data were deposited to the ProteomeXchange Consortium via the PRIDE (1) partner repository with the dataset identifier PXD016748.

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
