# Peer review of "SUMOylation Protects FASN Against Proteasomal Degradation in Breast Cancer Cells Treated with Grape Leaf Extract"

_biomolecules, 2020, doi:10.3390/biom10040529_

Round 1
Reviewer 1 Report
Major point
The main problem I see in this work has to do with its controls. The manuscript reads that “Untreated cells were used as a control”. For me, the control should be: that the cells were treated with the solutions of "hydroalcoholic extract of Vermentino" without the "leaf power" but the authors do not specify this and it is important to clarify it since, all the experimental results may change despite that it can be considered a minor detail.
Minor points
The authors refer to citation 41 for immunoprecipitation. This reference has nothing to do with the subject.
oligos for iRNA should be written from 5´ to 3´
Author Response
Reviewer 1
Major point
The main problem I see in this work has to do with its controls. The manuscript reads that “Untreated cells were used as a control”. For me, the control should be: that the cells were treated with the solutions of "hydroalcoholic extract of Vermentino" without the "leaf power" but the authors do not specify this and it is important to clarify it since, all the experimental results may change despite that it can be considered a minor detail.
We extensively specified it through the manuscript as reviewer suggested. Stimulated cells in the presence of the vehicle, water, were used as a control and named untreated because “leaf power” wasn’t added.
This is the procedure performed to derive the phenolic hydroalcoholic extracts we used for the treatments:
- we first reduced the vine leaves to fine powder using liquid nitrogen. We used liquid nitrogen for two reasons: 1) to avoid the oxidation of thermolabile compounds and to preserve the original characteristics of the leaves as much as possible; 2) to maximize the contact surface between the sample and the extracting hydroalcoholic solution. Although not specified in the manuscript, the extraction capacity of the hydroalcoholic solution is much greater on a reduced powder sample than on an intact leaf (this consideration is the result of our experience and our laboratory tests which are not mentioned in the manuscript because they would be redundant);
- once the leaf powder was obtained, it was put in contact with the hydroalcoholic extracting solution, for 16 hours, thus obtaining a suspension;
- after 16 hours, the suspension was centrifuged at 4629 g for 10 min at 4 °C (centrifuge A.L.C.-4227R, A.L.C. s.r.l., Milan, Italy). Only the supernatant, the extract containing the polyphenolic fraction, was frozen at -80 °C. At that temperature the water froze but the ethanol did not, it remained liquid and it was evaporated under a flow of gaseous nitrogen.
- Finally, the extract (now mainly aqueous) was subjected to lyophilization by freeze-drying (-60 °C). The lyophilization process allowed to eliminate all the water and any ethanol residues.
The result was the polyphenolic extract used for cell treatments, an ethanol-free extract powder. This extract was suitably diluted in the culture medium to carry out the treatments. That is why there is no ethanol-only control.
So, according to your suggestion, we modify the paragraph "Plant material and extraction procedure" as follow, so that the above is clear and there are no further misunderstandings:
Minor points
The authors refer to citation 41 for immunoprecipitation. This reference has nothing to do with the subject.
Reference chronology was modified including the missed reference mentioned above.
Oligos for iRNA should be written from 5´ to 3´
The oligo sequences for iRNA were indicated accordingly with the company datasheet.
Reviewer 2 Report
This is a very interesting paper and I congratulate the authors for their work. I have only a small number of comments and recommendations to make:
“Vitis vinifera” should be written in italics.
Line 110: “Phenolic compound was analyzed by LC-MS…”. The use of singular is not justified here. The extract contains a mixture of many (poly)phenolic compounds, therefore plural should be used here. The same holds true for lines 112-113.
Line 141: Please explain RIPA abbreviation (Radioimmunoprecipitation assay).
Line 175: “ The cells were then resuspended ...” This (in the context) suggests that they were resuspended after the flow cytrometry , whereas what follows shows that the resuspension was the first step in the flow cytometry analysis.
Line 193: it is not clear what reagents were used for the protein extraction.
Line 213: 1 × 106 ions is most likely 1x10^6. Same for line 215.
Line 225: PeptideProphet is a trade mark and should be written with appropriate capitals (P and P).
Line 246: According to the manufacturer’s info, Proteasome-Glo™ Assays are “not suitable for cell-based applications”, whereas the authors used cells for the assay. Are those assays valid if so?
The authors should justify the selection of the concentrations used.
Table 1 does not contain measurement units, neither is clear what spread measure is used (standard deviation, standard error etc).
The Materials and methods section lacks info on the statistical analyses (approaches, software, significance level etc).
With respect to Figure 5 and section “Vermentino hydroalcoholic extract lowers UBC9 protein level in human breast cancer cell lines”: the decrease of UBC9 in the SKBR-3 seems to be sizeably less pronounced than in MCF-7. A speculation on why this would happen might be useful for the readers.
Line 499: “unique” seems to be rather too strong here. Quercetin and isorhamntin glycosides (particularly the former, but also the latter) are not so rare in the plant world. Besides, the authors have not correlated the effect with these two flavonoids. Therefore I would rather recommend avoiding such a strong language for something that is rather speculative.
The authors use throughout the paper the word Vermentino to designate the herbal extract. In our view this is inappropriate, because they have provided no evidence that an extract of cv. Vermentino leaf is different from other Vitis vinifera L. leaf. Therefore, I recommend to use Vitis vinifera L. leaf instead of Vermentino.
Author Response
Reviewer 2
This is a very interesting paper and I congratulate the authors for their work. I have only a small number of comments and recommendations to make:
“Vitis vinifera” should be written in italics.
We modified it as reviewer suggested.
Line 110: “Phenolic compound was analyzed by LC-MS…”. The use of singular is not justified here. The extract contains a mixture of many (poly)phenolic compounds, therefore plural should be used here. The same holds true for lines 112-113.
We modified the line 110, 112 and 113 as reviewer suggested.
Line 141: Please explain RIPA abbreviation (Radioimmunoprecipitation assay).
RIPA abbreviation was already explained by the abbreviation list. The RIPA buffer composition was indicated as reviewer suggested.
Line 175: “ The cells were then resuspended ...” This (in the context) suggests that they were resuspended after the flow cytrometry , whereas what follows shows that the resuspension was the first step in the flow cytometry analysis.
The paragraph was revised as reviewer suggested.
Line 193: it is not clear what reagents were used for the protein extraction.
We modified the paragraph as reviewer suggested.
Line 213: 1 × 106 ions is most likely 1x10^6. Same for line 215.
We modified both lines as reviewer suggested.
Line 225: PeptideProphet is a trade mark and should be written with appropriate capitals (P and P).
We modified the line as reviewer suggested.
Line 246: According to the manufacturer’s info, Proteasome-Glo™ Assays are “not suitable for cell-based applications”, whereas the authors used cells for the assay. Are those assays valid if so?
Protesome-Glo assays has been demonstrated by us to be suitable for cell-base applications https://www.ncbi.nlm.nih.gov/pmc/articles/PMC2905543/pdf/nihms178056.pdf
The authors should justify the selection of the concentrations used.
We remarked that incubation with Vermentino leaf hydroalcoholic extract for 24 h significantly increased late apoptotic cells by 20% and necrosis by 25% compared to the control in both cell lines, with the greatest effect observed at a dosage of 400 μg/mL. This finding suggested us to select this dose for exploring the role of this extract on cell death regulation, where FASN plays a key role as negative regulator.
Table 1 does not contain measurement units, neither is clear what spread measure is used (standard deviation, standard error etc).
We corrected the caption as: Table 1 – Quantification (g*kg-1 ± standard deviation) of phenols in the hydroalcoholic extract of Vermentino grapevine leaves, obtained by HPLC-UV analysis.
The Materials and methods section lacks info on the statistical analyses (approaches, software, significance level etc).
We modified the line as reviewer suggested.
With respect to Figure 5 and section “Vermentino hydroalcoholic extract lowers UBC9 protein level in human breast cancer cell lines”: the decrease of UBC9 in the SKBR-3 seems to be sizeably less pronounced than in MCF-7. A speculation on why this would happen might be useful for the readers.
We hypothesized that the less pronounced effect on UBC9 protein level of Vermentino hydroalcoholic extract in the SKBR-3 occurs because SKBR-3 are HER2 positive breast cancer cells and well-known to be more aggressive compared to MCF-7 cells (https://www.ncbi.nlm.nih.gov/pmc/articles/PMC3782004/). In addition, Nieva and colleagues demonstrated that SKBR-3 show the highest variability in lipid metabolic genes expression and fatty acid composition suggesting their link with FASN role in breast cancer.(https://journals.plos.org/plosone/article?id=10.1371/journal.pone.0046456#authcontrib)
Line 499: “unique” seems to be rather too strong here. Quercetin and isorhamntin glycosides (particularly the former, but also the latter) are not so rare in the plant world. Besides, the authors have not correlated the effect with these two flavonoids. Therefore I would rather recommend avoiding such a strong language for something that is rather speculative.
We agree with the reviewer and changed the sentence as: “The phenolic profile obtained by HPLC-UV analysis suggests that the Vermentino leaf extract may have cytotoxic capability.”
The authors use throughout the paper the word Vermentino to designate the herbal extract. In our view this is inappropriate, because they have provided no evidence that an extract of cv. Vermentino leaf is different from other Vitis vinifera L. leaf. Therefore, I recommend to use Vitis vinifera L. leaf instead of Vermentino.
The reviewer notes an important aspect. He is right when he says that, in the paper, there is no evidence that the extract of cv Vermentino leaf is different from other Vitis vinifera L. leaves. In the manuscript the effect of hydroalcoholic extract of only one vine cultivar, Vermentino, was discussed and only the phenolic characterization of that specific variety is proposed. This, however, because the objective of the manuscript is not the comparison between extracts of leaves of different varieties.
It also does not mean that all vine varieties have the same phenolic profile. Indeed, the bibliography tells us the exact opposite. Please find here enclosed three different citations which demonstrate that different varieties have different phenolic profiles.
Loizzo et al., 2019. Comparative analysis of chemical composition, antioxidant and antiproliferative activities of Italian Vitis vinifera by-products for a sustainable agro-industry. Food and Chemical Toxicology 127, 127–134. doi: 10.1016/j.fct.2019.03.007
Monagas et al., 2006. Commercial Dietary Ingredients from Vitis vinifera L. Leaves and Grape Skins: Antioxidant and Chemical Characterization. J. Agric. Food Chem. 54, 319−327. doi: 10.1021/jf051807j
Šuković et al., 2020. Phenolic Profiles of Leaves, Grapes and Wine of Grapevine Variety Vranac (Vitis vinifera L.) from Montenegro. Foods, 9, 138. doi: 10.3390/foods9020138
For this reason, therefore, we think that not indicating the Vermentino variety can lead one to think that any Vitis vinifera leaves extract, of any variety, white or red, can have the same effect. This would be a generalization that could invalidate the reliability of future studies, leading to great confusion.

Round 2
Reviewer 1 Report
The authors have responded and promptly correctedthe observations presented.